# Treatment of Allergies to Fur Animals

**DOI:** 10.3390/ijms25137218

**Published:** 2024-06-29

**Authors:** Tomasz Rosada, Zbigniew Bartuzi, Magdalena Grześk-Kaczyńska, Magdalena Rydzyńska, Natalia Ukleja-Sokołowska

**Affiliations:** 1Chair and Clinic of Allergology, Clinical Immunology and Internal Diseases, Ludwik Rydygier Collegium Medicum in Bydgoszcz, Nicolaus Copernicus University in Toruń, 87-100 Toruń, Poland; medtom@op.pl (T.R.); zbartuzi@cm.umk.pl (Z.B.); 2Clinic of Allergology, Clinical Immunology and Internal Diseases, Jan Biziel University Hospital No. 2 in Bydgoszcz, Ujejskiego 75, 85-168 Bydgoszcz, Poland; magdalenagrzesk@gmail.com (M.G.-K.); magdalena_rydzynska@outlook.com (M.R.)

**Keywords:** allergy, fur animals, Fel d 1, treatment, therapy

## Abstract

Allergy to fur animals is becoming an increasingly common clinical problem in everyday medical practice. Depending on the route of exposure to the allergen, patients present with many, often non-specific symptoms. The most common illnesses among people with allergies to the above-mentioned allergens are as follows: allergic rhinitis, allergic conjunctivitis, atopic bronchial asthma, food allergy, allergic contact dermatitis, and sometimes anaphylactic shock. In recent years, there has been a change in the holistic approach to the treatment of allergy patients. The method of treatment should be tailored to a specific patient, taking into account his or her predispositions, economic possibilities, and therapeutic goals. The article describes the main methods of treating allergies, focusing primarily on allergies to fur animals. Allergy treatment always requires great care, and qualification for specific types of therapy should be preceded by a thorough and accurate diagnosis.

## 1. Introduction

Animal allergy is a clinical problem about which patients often consult with an allergologist. A study conducted on almost 13,000 German children and adolescents showed an allergic sensitization rate of 9.7% to dogs, 8.1% to cats, and 4.4% to horse hair [1], while in the Swedish BAMSE study, in a group of young adults, the frequency of allergic sensitization to cat allergens was 19.6%, to dog allergens 16.9%, and to horse allergens 9.8% [2]. There has been a noticeable increase in the frequency of allergic sensitization to fur animals along with the increasing age of the respondents, and in the group of adult Europeans reporting to a doctor with symptoms of inhalant allergy, approximately 26% are allergic to cat allergens, and 27% to dog allergens [3,4].

Table 1 below shows the allergens of the most popular fur animals, approved by the WHO/IUIS Allergen Nomenclature Sub-committee, including the individual protein families they belong to.

Depending on the route of exposure to the allergen, patients presenting numerous, often non-specific symptoms consult with a doctor. The vast majority of allergens from fur animals are classified as aeroallergens, but there are also food and contact allergens, which are mainly responsible for local skin reactions. The most common ailments among people with allergies to the above-mentioned allergens include the following: allergic rhinitis, allergic conjunctivitis, atopic bronchial asthma, food allergy, and allergic contact dermatitis [8,9,10].

In the case of allergy to fur animals, respiratory symptoms seem to be dominant, especially in adolescents and adults. Allergic rhinitis (AR) is usually a year-long/chronic ailment, which results from constant exposure to the allergen (e.g., dog owners, cat owners, horse stables workers, pig breeders, etc.). An interesting issue, from the point of view of people allergic to animal allergens, seems to be the so-called local allergic rhinitis, which is defined as the occurrence of typical symptoms of allergic rhinitis caused by an allergic reaction with the simultaneous absence of atopy in standard diagnostic tests [11,12]. In 2017, Hamizan et al. presented the results of a meta-analysis in which they assessed the incidence of local allergic rhinitis. The systematic review included 46 studies involving 3230 patients (1685 with AR and 380 with non-atopic rhinitis) and 165 healthy controls. It was found that the incidence of local allergic rhinitis in patients with non-allergic rhinitis was 24.7% (in the case of negative skin tests or no asIgE in serum) and 56.7% (in the case of negative skin tests and no asIgE) [13]. Such a situation can lead to considerable problems even for an experienced clinician, and currently the only form of diagnosis is a nasal provocation test, which is performed relatively rarely. Patients with local allergic rhinitis caused by animal allergens may not be correctly diagnosed, even with a seemingly obvious correlation: animal symptoms, which impede proper treatment and expose patients to further, burdensome diagnostics, do not allow the conscious elimination of the source of symptoms and maintain the pathomechanism of the disease through further exposure to the causative agent. Allergic bronchial asthma may also be the result of allergy to fur animals, and the connection between symptoms and exposure to the allergen is not always obvious, especially if the allergy does not concern common pets. Abrams et al. emphasize the possible involvement of house mouse allergens in the etiopathogenesis and course of allergic bronchial asthma. They report that exposure to the mouse allergen does not only concern people living in villages, but also occurs in the centers of large cities. The authors encourage mouse allergy to be taken into account during the diagnosis and treatment of patients with bronchial asthma, especially if the disease is not well controlled, despite a seemingly optimal treatment regimen [14].

It is worth emphasizing that food allergy caused by oral animal allergens most often occurs in early childhood, when it affects as many as 1.9% to 3.2% of children and is mainly related to exposure to cow’s milk allergens. Cow’s milk protein allergy is an extremely polysymptomatic disorder and its description is beyond the content of this article [15,16,17]. When discussing food allergy in terms of animal allergy, the possibility of clinical symptoms after eating meat cannot be overlooked. An extremely interesting fact is that not only the type of meat eaten, but also the fragment of the animal from which it comes can be important in the case of allergies. A good example is allergy to Bos d 13 (myosin light chain). So far, two variants of the myosin light chain have been characterized: the myosin 1 light chain (MYL1) and the myosin 3 light chain (MYL3). It has been shown that MYL1 is present primarily in fast-twitch fibers (type II), so they will dominate in such cuts of beef as ribs, pork neck, or rump steak, while MYL3 predominates in slow-twitch muscles (type I), so we expect them primarily in the tenderloin [18]. Cross-reactions may also be of potential importance, but a detailed discussion of the problem is beyond the scope of this article.

Anaphylactic shock after exposure to animal allergens is not a frequent phenomenon; nevertheless, the available literature describes cases of severe, acute allergic reactions, e.g., after exposure to allergens of horse milk, donkey milk, or sheep milk [19,20,21].

Severe allergic reactions to rodent bites, such as the Djungarian hamster, mouse, and rats, have also been reported, but such cases appear to be incidental [22,23].

While diagnosing allergies to fur animals, we can use a whole arsenal of allergy tests, but not all of them have the same sensitivity and specificity. What is more, in some cases, their limited availability may also be a problem. The most frequently used tests are skin prick tests, allergen-specific IgE (asIgE) tests using allergen extracts and specific allergen components, as well as provocation tests, although the latter are performed only occasionally. In exceptional cases, the latest diagnostic methods can be used, such as the basophil activation test or the mast cell activation test, often called a test tube challenge test. It is worth emphasizing that the diagnosis of allergy to fur animals is not easy and requires clinicians to have extensive knowledge and exercise caution in taking therapeutic decisions as well as creating appropriate recommendations for the patient [4,24,25]. A detailed discussion of the diagnosis of fur animal allergy is presented in the article “*Diagnostics of Allergy to Furry Animals-Possibilities in 2024*” published in the *Journal of Clinical Medicine*, which was prepared by some of authors of this publication [26].

## 2. Allergy Treatment

In recent years, there has been a change in the holistic approach to the treatment of allergy patients. The most significant assumption of the treatment is a more patient-centered therapy, with an emphasis on personalized, predictive, preventive, and participatory strategies. The method of treatment should therefore be tailored to a particular patient, taking into account their allergy profile, the anticipated course of allergy, economic possibilities, and individualized therapeutic goals. While planning the therapy, it is advisable to predict the possible consequences of the pathomechanism of the disease itself, as well as the possible consequences of the undertaken therapeutic activities, in order to prevent subsequent adverse health effects for the patient. The patient’s involvement in the treatment process is very important, because the proper therapeutic compliance is a basic component of the doctor–patient relationship and is the foundation for the success of therapy [27,28,29,30].

The current therapeutic possibilities for allergic diseases are presented in Figure 1. Below, there is also a brief description of them, including the main assumptions and the most significant limitations.

Eliminating the allergen responsible for the development of clinical symptoms, from the point of view of the pathomechanism of allergy, appears to be the ideal and simplest solution to the problem. No allergen, no symptoms. However, it is not always possible to implement in practice. The routes of spreading and the nature of allergens (inhalant, food, contact allergens) have a significant impact on the possibility of the selective removal of the causative agent. Eliminating the allergen in the context of food allergy seems to be the easiest and most effective treatment option. At this point, however, attention should be paid to the numerous limitations of this method, such as the possibility of cross-allergies, the problem of the so-called “hidden allergens”, or the lack of the possibility of using adequate substitutes to maintain the patient’s healthy and balanced diet. Inhalant allergens, on the other hand, both seasonal and year-round ones, are practically impossible to eliminate from the patient’s environment, but the exposure can be reduced. In the case of dog or cat fur, exposure can be limited by removing these pets from the patient’s nearest environment. However, it happens that in severely allergic people, even a small amount of animal allergens, e.g., on the clothes of a household member, may be sufficient to induce clinical symptoms, including the exacerbation of bronchial asthma. In addition, a lot of people who are allergic to dog or cat allergens want to own such a pet. This makes it necessary to look for other therapeutic options [31,32,33,34].

Specific immunotherapy is another possibility for the specific treatment of allergic diseases. Its main assumption is giving the allergen responsible for causing clinical symptoms to the patient in regular, increasing doses in order to induce tolerance [35,36]. It is currently thought that the mechanism of action of immunotherapy is primarily based on two main phenomena, i.e., the change in immune polarization/deviation and the induction of regulatory T cells. During immunotherapy, a single dose of the allergen is relatively higher than the estimated maximum annual exposure and is delivered via a different route than during the natural induction of allergic processes, i.e., subcutaneously or sublingually. It results in a specific retuning of immunological processes and the gradual development of tolerance to a given allergen. The change in immune polarization consists in redirecting the allergic reaction towards T1-dependent inflammation at the expense of T2 inflammation, i.e., Th1 lymphocytes are mobilized and stimulated under the influence of the supplied allergen and secondarily they secrete gamma interferon (IFN-γ), stimulating B lymphocytes to produce IgG instead of IgE. Immunoglobulins G, in turn, are unable to trigger an allergic reaction, so the disease process is partially inhibited. The second mechanism is based on inducing allergen-specific CD4^+^CD25^+^ regulatory lymphocytes (asTreg), producing IL-10 and TGF-β and suppressing the local Th2-dependent response. IL-10 and TGF-β redirect the antibody class-switching in favor of IgG4 and IgA, which stop the presentation of the allergen to Th2 cells and additionally block the allergen-induced activation of mast cells and basophils, thereby significantly weakening the allergic reaction [37,38]. The main contraindications to specific immunotherapy are as follows: age < 2 years, uncontrolled bronchial asthma, multiple-organ autoimmune diseases or uncontrolled psoriasis, and malignant tumors. The implementation of immunotherapy in treatment offers new opportunities to significantly improve the quality of life of patients suffering from allergic diseases, but in many cases, it still requires further clinical trials [39,40].

In pharmacological treatment in allergology, various groups of drugs are used in order to suppress symptoms or slow down the progression of the disease, mainly by interfering in the pathomechanism of the disease. Table 2 shows the main groups of drugs used in the treatment of allergic diseases, taking into account their mechanism of action.

## 3. Treatment of Allergies to Fur Animals

### 3.1. Prevention of Allergies to Fur Animals

The term prevention applies to a very wide range of factors, modifiable and independent, which cannot be eliminated, but which have a potentially huge influence on the immune balance and possible predispositions to the development of allergies. Most often, there are four stages of prevention, which, however, interpenetrate in many aspects and include subsequent stages of the pathomechanism of the disease [44]. The goal of early prevention is to consolidate correct patterns of a healthy lifestyle and prevent the spread of unfavorable behavioral patterns among healthy people. In primary prevention, we attempt to prevent the disease by controlling risk factors in people exposed to these factors, whereas secondary prevention focuses primarily on preventing the consequences of the disease through its early diagnosis and treatment. The last stage includes activities connected with tertiary prevention, which involve the attempts to stop the progression of the disease and limit its complications [45].

In 2022, “S3 Guideline Allergy Prevention” was published in *Allergologie Select*. The authors, based on an in-depth analysis of the latest scientific reports, created recommendations which, in light of the limited possibilities of causal treatment, are supposed to improve the primary prevention of bronchial asthma, allergic rhinitis, food allergy, and atopic dermatitis. The recommendations include allergy risk factors occurring as early as in fetal life and provide an opportunity to reduce the spread of various allergic diseases. In accordance with the above-mentioned guidelines, dietary restrictions should not be introduced during pregnancy or lactation in order to prevent allergies, and after birth, infants should be only breastfed for a period of 4–6 months. Importantly, breastfeeding, if possible, should be continued even after the introduction of the first complementary foods, and cow’s milk-based formulas should be avoided, especially in the first days of the child’s life. If breastfeeding is not possible, the evaluation of individual allergy risk factors is advisable, and in the case of infants from the high-risk group, it is recommended to use special formulas with proven effectiveness in reducing the risk of allergy. Within primary prevention, it is not recommended to give the child formulas based on the milk of other animals, such as goat, sheep, or mare, and soy milk can be used as complementary food, regardless of the purpose of allergy prevention. There is no evidence of the preventive effect of dietary restrictions such as avoiding strong food allergens in the first year of life; therefore, no restrictions should be introduced. Women planning pregnancy, as well as children and adolescents, should maintain a proper body weight (proper BMI), as it has been unequivocally proven that a mother’s excess weight during pregnancy as well as obesity in childhood closely correlate with episodes of wheezing and the development of bronchial asthma later in life. Currently, studies have not proven a protective role of vitamin D, other fat-soluble vitamins, or omega acids in the development of allergic diseases in any study group. All vaccinations should be carried out in accordance with the vaccination schedule, since they do not increase the risk of developing allergies. Children growing up in rural areas have a slightly lower risk of developing asthma and other allergic diseases, which is probably due to early, non-specific immunostimulation by the microbiological composition of house dust. Exposure to tobacco smoke (concerning both active and passive smoking) and particulate matter (PM) 2.5 (atmospheric aerosols whose diameter is not bigger than 2.5 μm) have proven pro-allergic properties at all ages, starting as early as in fetal life [46]. The above recommendations are consistent with the German guidelines, published in 2010 in *Journal of the German Society of Dermatology*, and the 2014 “S3 Allergy Guidelines”, published in the *Allergo Journal International,* which highlight the relevance of the issues raised and grounds them in proper clinical management [47,48].

We can also talk about prevention in terms of preventing the development of allergies in workers who are in contact with laboratory animals. It is estimated that one in three of the above-mentioned employees may develop disease symptoms resulting from an allergy to laboratory animals. Most people develop allergic conjunctivitis, urticaria, or bronchial asthma, but anaphylactic shock has also been reported after being bitten by a mouse or rat, and after being pricked by a needle used when working with a rabbit [49]. Among employees who were not provided with any preventive measures, allergic symptoms within the first year of work occurred in 5–40% of those exposed. IgE-dependent allergy, based on skin tests or the determination of specific IgE in serum, was found in over 60% of those presenting clinical symptoms. The studies conducted so far have also proven that people with atopy are 11 times more likely to be allergic to laboratory animal allergens [50]. The significance of the problem is emphasized by recommendations regarding prevention issued, among others, by the National Institute for Occupational Safety and Health (NIOSH) in the United States or the Guidelines of the Health and Safety Executive (HSE) in the United Kingdom, as well as the inspections conducted among laboratory employees [51]. It has been proven that the development of allergy to laboratory animals is most influenced by the exposure to allergens, and even low levels of exposure may pose a risk of developing the disease [52].

Many papers have been published addressing the issue of the exposure/elimination of animal allergens as a major risk factor for the development of fur animal allergies. Epidemiological studies have shown that having a dog in the home during the first three years of a child’s life has a protective effect, primarily in relation to the development of food allergies and allergic bronchial asthma, until the age of 13 [53,54,55,56]. Due to the discrepancies obtained in the results of the conducted research, having cats and other typical pets cannot be clearly defined as positive or negative in terms of allergy prevention [46].

Additional variables that may directly and indirectly affect the results of research about the role of exposure to animal allergens in the pathomechanism of allergic diseases increase the potential difficulties in obtaining clear answers to the questions asked. In their research results, Almqvist et al. noticed that cats were less frequently kept in families where the parents had asthma, rhinoconjunctivitis, and allergies to pets or pollen (3.5–5.8%) than in families where the parents did not suffer from an allergic disease (10.8–11.8%). Dogs, on the other hand, were less common in families where atopic dermatitis was diagnosed (3.3%) than in the families where this disease was not observed (5.9%) [57]. Thus, a family history of allergies may influence the risk of developing allergies in a child, not only through genetic predisposition, but also through the avoidance of particular allergens, because of the focus on the well-being of other household members (parents). In his publication, Morris notices that the risk of allergy and the onset of symptoms in childhood (age ≥ 3 years) seems to depend on the mother’s history of allergies (mainly bronchial asthma). Children of mothers with asthma had an increased risk of sensitization to Fel d 1 with high exposure, whereas in children whose mothers did not suffer from asthma, exposure to high concentrations of Fel d 1 appeared to have a protective effect [58]. In 2000, *The Lancet* published the results of a study which evaluated the exposure to allergens and the concentration of sIgE directed against house dust mite and cat allergens in 1314 newborns. The assessment was made at four time points, i.e., at birth, at 18 months, at 3 years, and in the 7th year of the child’s life. The complete data were obtained for over 71% of originally enrolled participants. It was found that children who suffered from wheezing (10%) or were diagnosed with bronchial asthma (6.1%) were significantly more likely to have sIgE to the inhalant allergens tested; however, no relationship was confirmed between the exposure to inhalant allergens and the incidence of bronchial asthma [59]. To sum up, it should be noted that induced tolerance to fur animal allergens in children has the potential to persist into adolescence and adulthood [58]. According to current knowledge, in families where there is no significantly increased risk of allergy, cats and dogs should not be eliminated from the environment due to the primary allergy prevention. However, in relation to pets other than cats and dogs, no recommendations can be made regarding the primary prevention of allergic diseases [46,47,48].

### 3.2. Elimination of the Causative “Agent”

When treating an allergy to fur animals, the easiest solution seems to be the elimination of the causative “agent”. However, as mentioned earlier, removing the animal from the patient’s immediate environment does not always bring the intended result, since allergens can also be delivered via indirect routes, such as by household members or co-workers. The presence of animal allergens was shown in dust samples taken from places where the animal had never been. It is reported that the amount of cat allergens sufficient to develop allergies and clinical symptoms persists up to 10 years after the animal is removed from the home [60,61,62].

For episodic or occupational exposure, direct barrier methods may turn out to be effective. Their use makes it possible to limit the number of allergens delivered to the body, and thus minimize the intensity of the disease symptoms. People allergic to fur animal allergens should prepare appropriately before an expected exposure to them. It may be advisable to minimize the time of exposure to the allergen, ventilate the rooms in which the allergy sufferer will stay, avoid direct contact with the animal, e.g., by placing the animal in another room, and using protective masks, disposable gloves, or other protective measures for the avoidance of direct contact with the allergen. After returning home, the allergic person should rinse the mouth, clean the nasal passages, change clothes, which should be washed, e.g., with the addition of special cleaning agents, and then thoroughly wash the entire body, including the hair, to remove even the smallest amounts of allergenic particles. Antihistamines can also be considered in preparation for contact with the allergen [63,64,65].

A practical problem in the aspect of eliminating the causative “agent” in the case of animals is the emotional sphere. In their study, Janssens et al. showed that the presence of companion animals and interaction with them is linked to aspects of the emotional well-being of their owners. The presence of a companion animal can protect people against negative feelings, while interacting with it can generate positive emotions [66]. Based on a survey, Martens et al. found that all pet owners showed strong attachment to their companion animals, with the degree of attachment (for both cat and dog owners) varying significantly, depending on education level and gender [67]. The beneficial effect of animals on humans is not only one-way, as confirmed by Rault et al. in their publication emphasizing that positive experiences with people lead to pets looking for people and willingly interacting with them, which they treat as an intrinsic reward [68]. Therefore, a simple “elimination” of an allergenic factor such as an animal can have a huge impact on the psychosomatic condition of a person and provoke severe stress, which is defined as an imbalance between the body’s requirements and the body’s capabilities to meet them. Also, stress has a major impact on the human immune system. The activation of the neuroendocrine and sympathetic systems through the secretion of catecholamines and cortisol influences the immune system, modifying the balance between the Th1/Th2 response in favor of Th2 action. It cannot be unequivocally stated that chronic stress is itself capable of causing allergy, although evidence from various studies suggests that stress in genetically susceptible people may, on the one hand, promote the occurrence of an allergic disease and, on the other hand, make it difficult to control an already existing one [69].

### 3.3. Specific Immunotherapy

Specific immunotherapy is also used to treat allergies to fur animals. Currently, it is possible to administer allergens subcutaneously (subcutaneous immunotherapy, SCIT) or sublingually (sublingual immunotherapy, SLIT). Both methods allow achieving the main goal, inducing immunological tolerance and reducing clinical symptoms, but the mechanism of this induction is not the same. Scientific research is also attempting to use other routes of allergen administration, such as intralymphatic injection (intralymphatic immunotherapy, ILIT), and the results of these studies appear to be very promising [70,71]. When qualifying a patient for a specific immunotherapy, it is fundamental to make the correct diagnosis and establish the relationship between clinical symptoms and the confirmation of IgE-dependent allergy with diagnostic tests. Currently, the most popular tests in this respect are skin prick tests with allergen extracts and the determination of specific IgE. However, taking into account the possibility of cross-reactions and the frequent polysensitization of patients, it would be worthwhile, during qualification, to evaluate the entire individual allergic profile of the patient by performing, for example, molecular tests, and in some cases also inhibition tests, which are still an experimental method. Correct qualification has a huge impact on the success of this type of treatment. Importantly, in the light of the current guidelines, the use of a specific immunotherapy should be considered in patients whose clinical symptoms remain uncontrolled in spite of appropriate pharmacological treatment and the avoidance of allergens whenever possible. In the case of allergy to fur animals, immunotherapy is most often considered when exposure to animal allergens cannot be prevented, for example, due to the patient’s profession (veterinarian, policeman, firefighter, etc.) or environmental circumstances [72,73].

The results of the studies on the effectiveness of specific immunotherapy in the treatment of allergies to fur animals provide inconclusive results. Chu et al. report that AIT using cat and/or dog hair may be an effective therapeutic option in patients with atopic dermatitis (AD), especially in patients with severe AD and other allergic respiratory diseases when exposure to animal hair cannot be avoided. It is worth noticing, however, that this study was conducted on a group of only 19 patients [74]. In turn, Senti et al. attempted to evaluate the effectiveness of intralymphatic immunotherapy using MAT-Fel d 1 (a modular vaccine transporting the Fel d 1 antigen). No adverse events were reported during the study. Three intralymphatic injections of MAT-Fel d 1 increased nasal tolerance 74-fold (*p* < 0.001 compared with placebo). ILIT with MAT–Fel d 1 stimulated the regulatory T cell response (*p* = 0.026 vs. placebo) and increased the level of Fel d 1-specific IgG4 5.66-fold (*p* = 0.003). The IgG4 response was positively correlated with IL-10 production (*p* < 0.001) [75]. These optimistic results stand in opposition to the findings published by Smith and Coop on AIT for dog allergens. Based on the analysis of numerous scientific reports, they concluded that immunotherapy with dog hair extracts in patients with dog hypersensitivity shows poor and contradictory results of clinical effectiveness, which is attributed to the low quality of the extracts and the innate, complex allergic profile of dogs, which lacks a clearly dominant allergen [76]. Colque-Bayona M. et al. came to different conclusions. They assessed the effect of subcutaneous immunotherapy in patients who are allergic to dog and cat allergens on the patients’ quality of life, focusing on clinical symptoms, the need to use antiallergic medications, the number of bronchial asthma exacerbations, and immunological factors. They showed statistically significant improvement in reducing nasal symptoms, lowering the number of asthma exacerbations, and a decrease in the amount of anti-allergic drugs used. However, it should be noted that only 13 patients were included in the study and the entire study was defined as a pilot study [77].

Publications on AIT in groups of patients who are allergic to fur animals focus mainly on cats and dogs. The available literature lacks data on other fur animals, such as hamsters, mice, or rats.

A summary of allergen extracts available in Poland based on the Register of Medicinal Products [78] is presented below in Table 3.

### 3.4. Pharmacological Treatment

The pharmacological treatment of animal allergies does not differ fundamentally from the treatment of allergies to other airborne allergens. It depends largely on the clinical picture and the course of allergy. It is primarily symptomatic in nature. Most often, an allergy to animals manifests itself as skin lesions such as urticaria, chronic allergic rhinitis, obstructive dyspnea, and less often, the allergy may have a severe, life-threatening course, including anaphylactic shock. The treatment of individual diseases should be consistent with the existing guidelines, including those of the Polish Society of Allergology and the European Academy of Allergology and Clinical Immunology [79,80].

In general, it can be assumed that in the case of chronic exposure to an animal allergen and the resulting atopic disease, treatment should be chosen individually, depending on the clinical course, aimed at reducing the symptoms, providing comfort to the patient, and preventing the progression of the allergic disease [81,82]. Currently, no specific guidelines have been developed for the treatment of animal allergies, but based on numerous scientific publications and available recommendations concerning pharmacotherapy in allergic diseases, it is possible to attempt to adapt them to the needs of patients who are allergic to animal allergens [79,80].

Oral antihistamines are mainly used in the treatment of allergic rhinitis, allergic conjunctivitis, and urticaria. Currently, second-generation antihistamines are preferred, primarily due to the lack of a sedative effect and lower anticholinergic effect, which may constitute a significant clinical problem, especially in elderly groups, putting them at risk of a dry mouth, urinary retention, or delirium [83,84]. In the case of animal allergies, two treatment options should be considered: acute and chronic.

In people who are allergic to animal allergens but do not come into direct contact with animals on a daily basis, it may be advisable to provide them with a fast-acting antihistamine drug and train them so that they can appropriately use the recommended preparation in the event of the exposure to a specific allergen. In this aspect, rupatidine and bilastine seem to be the best choices, mainly due to the rapid onset of action; rupatidine starts working as soon as about 30 min after taking it, while bilastine starts working after about 1 h. Both drugs have a very good safety profile [85,86]. The dose of antihistamine should be selected individually, depending on the frequency of the contact with animals and the severity of clinical symptoms. In some situations, a double or even triple dose may be necessary to use [87].

In the case of chronic exposure to animal allergens, the chronic use of antihistamines may turn out to be necessary. In such a situation, it is advisable to use the lowest effective dose of the drug, with a possible increase in the dose in the event of higher exposure to the sensitizing allergen or the intensification of clinical symptoms. All second-generation antihistamines have a similar duration of action, i.e., approximately 24 h; for bilastine, the duration of action is said to be in the range of 24–26 h, which may be of minor importance, e.g., if one dose of the drug is missed [85,86,88]. It has been proven that the long-term use of antihistamines does not reduce their effectiveness; therefore, it is not recommended to replace antihistamines in order to obtain a better effect. In addition, recommendations regarding the possible rotation of the drug within a given group have not yet been found in any of the stances of the main societies providing advice on the use of antihistamines [89].

If the clinical symptoms are limited to only one organ, e.g., allergic conjunctivitis, the first choice should be topical antihistamines in the form of drops or intranasal preparations. This approach seems to be optimal approach because the drug immediately reaches the site of the ongoing allergic reaction, while the systemic effect of the active substance used is limited. Moreover, if a topical drug allows for a reduction in the dose of an oral drug, such a combination is preferred because it reduces the severity of possible side effects and improves the safety of chronic therapy [90,91].

If bronchial asthma develops due to an animal allergy, treatment should be carried out in accordance with the current guidelines. There are no recommendations regarding differences in asthma therapy depending on the allergic profile of the patient—with regard to animal allergies [79,80].

#### Biological Treatment

Biological treatment is another breakthrough in the allergy therapy. Currently, an increasing number of drugs are used, with great effectiveness and a high safety profile. In Poland, they are available mainly within so-called drug programs. The main biological drugs used in the treatment of allergic diseases are presented in Table 4. Biological therapy is dedicated to a specific disease entity. In some cases, fur animals can be considered a year-round allergen, causing chronic symptoms. In that case, allergy to them can be considered an eligibility criterion for biological treatment with omalizumab and dupilumab in the case of severe atopic bronchial asthma.

However, it cannot be ruled out that in terms of animal allergies, treatment with all the biological drugs listed in Table 4, apart from lanadelumab, may be effective.

### 3.5. New Perspectives in the Treatment of Allergies to Fur Animals

The increasing prevalence of animal allergies, as well as the desire to own a pet, and the refusal of patients who are allergic to animal allergens to get rid of the source of the allergens, create the need to look for new ways of treating this type of allergy.

In the available literature, we can encounter the concept of hypoallergenic animals, but Hilger et al. emphasize in their publication that molecular analyses of animal dander from different breeds show high individual variation in allergen levels, but do not provide any scientific evidence for the concept of hypoallergenic cats, dogs, or horses [92].

In 2019, Thoms et al. presented an innovative idea for treating cat allergy, specifically to one of its main allergens—Fel d 1. The entire intervention focused on the “source of allergy”, in this case the cat. The researchers immunized cats, sensitizing them to their own protein, uteroglobin, by administering recombinant Fel d 1 in combination with a virus-like particle that is an epitope for T lymphocytes. The vaccine was very well tolerated and had no overt adverse effects. In all cats undergoing the procedure, anti-Fel d 1 antibodies in the IgG class were successfully induced, which were characterized by high affinity and showed strong neutralizing ability; the vaccine was tested both in vitro and in vivo, but most importantly, a significant decrease in the level of endogenous allergens in the environment of the tested cats was observed [93].

Nowadays, allergological diagnostics increasingly often attempts to determine the patient’s individual molecular allergen profile, which creates new opportunities for the treatment of allergic diseases. Shamji et al. developed two monoclonal IgG antibodies against Fel d 1 (REGN1908-1909) that simultaneously and non-competitively bound to conformational epitopes of the major cat allergen. In a randomized, double-blind, placebo-controlled study, they evaluated the effectiveness of a single, subcutaneous dose of REGN1908-1909 in patients who were allergic to cats. Intranasal provocation with cat allergens was performed on the day of the drug administration and then on the 8th, 29th, 57th, and 85th days. The treatment provided a significant reduction in the intensity of nasal symptoms at all control points except day 57. The study also showed that REGN1908-1909 was well tolerated. The main benefit of the procedure was the rapid response to therapy, with benefits observed after a few days of treatment rather than after many months, as in the case of traditional allergen immunotherapy. The study results create an opportunity for new, effective allergy treatment methods that would be precisely tailored to the patient’s individual allergic profile and would allow interference with the immune system only to the extent necessary to alleviate the disease symptoms [94].

Given the unclear role of Fel d 1 in the functioning of cats, efforts have been made to develop a method of eliminating this protein, but only after transferring it to the animal’s fur and still before inducing the disease symptoms in an allergic human. Satyaraj et al. assessed the effectiveness of Fel d 1 binding by immunoglobulin Y derived from egg yolk. It has been shown that anti-Fel d 1 IgY can be induced by exposing chickens to Fel d 1, and when added to cat saliva, it effectively blocks the binding of IgE to Fel d 1 in a dose-dependent manner [95]. In the next stage of the study, saliva was collected from twenty healthy adult short-haired domestic cats 5 h after morning feeding, 5 days a week, for the study period of 5 weeks. Cats were fed a control diet for 1 week of the baseline period, followed by a control diet (control group) or a control diet with an egg product containing anti-Fel d 1 IgY (test group) for 4 weeks. Salivary Fel d 1 levels were significantly reduced by week 3 in cats receiving anti-Fel d 1 IgY in the diet, by an average of 24%, while the control group showed no significant reduction in active Fel d 1. In the final stage, the effect of the intervention on the ability to reduce clinical symptoms in allergic people was tested. Eight short-haired domestic cats were selected and divided into two equal groups. Then, for a period of 8 weeks, the cats were fed a test diet, and the test diet in the test group was supplemented with anti-Fel d 1 IgY. During the last 4 weeks of the study, blankets were placed in the room used by each group of cats, which were used by the animals as bedding. After this time, the blankets were used in the provocation of the selected people who were allergic to Fel d 1. The provocation was continued for 3 h or until the disease symptoms in the subjects were evaluated as unbearable. The patients recorded their symptoms and their severity on Total Nasal Symptom Score (TNSS) and Total Ocular Symptom Score (TOSS) sheets every 15 min throughout the procedure. Despite the small study group (11 people), this study showed that feeding cats a diet containing anti-Fel d 1 IgY antibodies reduces Fel d 1 levels in the cat’s environment and leads to significant improvements in the total nasal symptom score and some ocular symptoms in patients allergic to cats [96]. Currently, cat food containing anti-Fel d 1 IgY is commercially available and is advertised as food for cats that reduces their allergenicity [97].

## 4. Conclusions

Allergy treatment requires great care, and qualification for specific types of therapy should always be preceded by a thorough and accurate diagnosis. The basic treatment regimen for animal allergies does not differ significantly from other types of allergies, but a lot of studies are currently being conducted on new methods of therapy that may bring new solutions and improve the quality of life of patients.

## Figures and Tables

**Figure 1 ijms-25-07218-f001:**
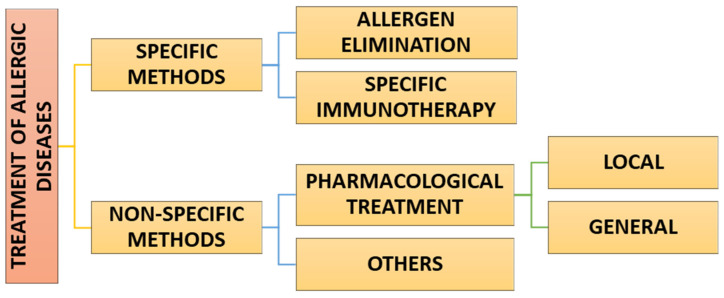
Main therapeutic methods in allergology.

**Table 1 ijms-25-07218-t001:** Allergens of fur animals, taking into account particular protein families they belong to [4,5,6,7].

Animal	Dog	Cat	Guinea Pig	Djungarian Hamster	Golden Hamster	House Mouse	Sewer Rat	European Rabbit	Domestic Cattle	Horse	Common Donkey	Domestic Pig
Allergens	Can f 1	Fel d 1	Cav p 1	Phod s 1	Mes a 1	Mus m 1	Rat n 1	Ory c 1	Bos d 2	Equ c 1	Equ a 6	Sus s 1
	Can f 2	Fel d 2	Cav p 2					Ory c 2	Bos d 3	Equ c 2		
	Can f 3	Fel d 3	Cav p 3					Ory c 3	Bos d 4	Equ c 3		
	Can f 4	Fel d 4	Cav p 4					Ory c 4	Bos d 5	Equ c 4		**LEGEND**
	Can f 5	Fel d 5	Cav p 6						Bos d 6	Equ c 6		lipocalin
	Can f 6	Fel d 6							Bos d 7			serum albumin
	Can f 7	Fel d 7							Bos d 8			kallikrein
	Can f 8	Fel d 8							Bos d 9			cystatin
									Bos d 10			secretoglobin
									Bos d 11			immunoglobulin
									Bos d 12			casein
									Bos d 13			lysozyme
												others

**Table 2 ijms-25-07218-t002:** Main groups of drugs used in the treatment of allergic diseases [41,42,43]. The importance of individual groups of drugs in relation to allergy to fur animals is presented in Section 3.4: Pharmacological Treatment.

Group of Drugs	Examples of Substances	Mechanism of Action	Comments	Importance in Treating Allergies to Fur Animals
First-Generation Antihistamines	Hydroxyzine Prometazine Clemastine	H1 receptor antagonist (a special type of antagonist with so-called inverse agonism).		Treatment of allergic rhinitis, allergic conjunctivitis, and urticaria (allergy to fur animals). Prevention against episodic exposure to fur allergens. Emergency treatment in case of unexpected exposure to fur allergens. Chronic treatment if necessary.
Second-Generation Antihistamines	Bilastine Cetirizine Loratidine	H1 receptor antagonist (a special type of antagonist with so-called inverse agonism).	More selective and longer-lasting effect on peripheral H1 receptors. Less impact on H1 receptors in the CNS.	Treatment of allergic rhinitis, allergic conjunctivitis, and urticaria (allergy to fur animals). Prevention against episodic exposure to fur allergens. Emergency treatment in case of unexpected exposure to fur allergens. Chronic treatment if necessary.
Glucocorticosteroids	Prednisone Prednisolone Dexamethasone	Multidirectional anti-inflammatory effect. Receptor agonists for natural glucocorticoids.	Used systemically, via inhalation, intranasally to conjunctival sac, and externally.	Local treatment of allergic rhinitis, allergic conjunctivitis, and atopic dermatitis (allergy to fur animals). Inhalation treatment in allergic bronchial asthma (allergy to fur animals). Systemic treatment in severe exacerbations of allergic bronchial asthma or atopic dermatitis (allergy to fur animals). Emergency treatment of acute allergic reactions following exposure to fur allergens, if necessary.
Antileukotriene Drugs	Montelukast Zafirlukast Zileuton Weliflapon Atuliflapon	1. Cysteinyl leukotriene receptor antagonist (montelukast, zafirlukast). 2. 5-LOX inhibitor (zileuton) 3. 5-LOX activating protein inhibitor (veliflapon, atuliflapon).		Supportive treatment of allergic bronchial asthma.
β2-Mimetic Drugs	Phenotherol Salbutamol Salmeterol Formoterol	β2 receptor agonist.		Inhalation treatment of allergic bronchial asthma.
Anticholinergic Drugs	Ipratropium Tiotropium Glycopyronium Umeclidinium	Muscarinic receptor antagonist.		Supportive treatment of allergic bronchial asthma.
Immunosuppressive Drugs	Cyclosporine Everolimus Methotrexate Tacrolimus	Depending on a particular drug.		Severe atopic dermatitis, if necessary.
Biological Drugs	Omalizumab Mepolizumab Roslizumab Benralizumab Dupilumab Tezepelumab Lanadelumab	Depending on the specific molecule.	Available only as a part of so-called drug programs.	
Methylxanthines	Theophylline	A weak, non-selective phosphodiesterase inhibitor.		Supportive treatment of allergic bronchial asthma.
Chromones	Disodium Cromoglycate	Stabilization of the mast cell membrane and inhibition of the release of pro-inflammatory mediators.		Supportive treatment of allergic bronchial asthma.
Magnesium Sulfate		Modulation of calcium channel activities.		Supportive treatment of allergic bronchial asthma.

**Table 3 ijms-25-07218-t003:** List of animal allergen extracts for specific immunotherapy [78].

Trading Name	Producer	Manufacturer Code	Allergen Extract	Route of Administration
Novo-Helisen Depot	Allergopharma GmbH & Co., KG (Reinbek, Germany)	304	Hamster fur	Subcutaneously
		306	Dog fur	Subcutaneously
		308	Rabbit fur	Subcutaneously
		309	Cat fur	Subcutaneously
		311	Guinea pig fur	Subcutaneously
		314	Horse fur	Subcutaneously
		317	Cow fur	Subcutaneously
		318	Sheep wool	Subcutaneously
		321	Parrot feathers	Subcutaneously
Alutard SQ	ALK-Abelló A/S (Hørsholm, Denmark)	552	Horse fur	Subcutaneously
		553	Dog fur	Subcutaneously
		555	Cat fur	Subcutaneously
Staloral	STALLERGENES (Baar, Switzerland)	507	Cat	Orally/sublingually
		509	Dog	Orally/sublingually
		508	Goat	Orally/sublingually
		510	Guinea pig	Orally/sublingually
		511	Hamster	Orally/sublingually
		516	Horse	Orally/sublingually
		512	Rabbit	Orally/sublingually
		505	Sheep wool	Orally/sublingually
Phostal	STALLERGENES	507	Cat	Subcutaneously
		509	Dog	Subcutaneously
		508	Goat	Subcutaneously
		510	Guinea pig	Subcutaneously
		511	Hamster	Subcutaneously
		516	Horse	Subcutaneously
		512	Rabbit	Subcutaneously
		505	Sheep wool	Subcutaneously

**Table 4 ijms-25-07218-t004:** Major biologic drugs used in allergology [42].

Biological Drug	Mechanism of Action	Indications	Importance In Treating Allergies to Fur Animals
Omalizumab	Anti-IgE antibody	Severe allergic asthma Severe chronic rhinosinusitis with polyps Chronic, refractory spontaneous urticaria	Treatment of allergic bronchial asthma associated with allergy to fur allergens.It is worth noting that one of the qualifying criteria for the treatment of bronchial asthma with omalizumab is year-round exposure to the allergen (fur animal allergens are classified as year-round if we cannot eliminate the exposure).
Mepolizumab	Anti-IL-5 antibody	Severe eosinophilic asthma Chronic rhinosinusitis or sinusitis with polyps Eosinophilic granulomatosis with polyangitis Idiopathic hypereosinophilic syndrome	Treatment of allergic bronchial asthma associated with allergy to fur allergens.
Reslizumab	Anti-IL-5 antibody	Severe eosinophilic asthma	Treatment of allergic bronchial asthma associated with allergy to fur allergens.
Benralizumab	Anti-IL-5Rα antibody	Severe eosinophilic asthma	Treatment of allergic bronchial asthma associated with allergy to fur allergens.
Dupilumab	Anti-IL-4R/anti-IL-13R antibody	Moderate to severe atopic dermatitis Adjunctive maintenance treatment for severe asthma with type 2 inflammation Adjunctive treatment in combination with intranasal corticosteroids for severe chronic rhinosinusitis with nasal polyps Prurigo nodularis Eosinophilic esophagitis	Treatment of atopic dermatitis or allergic bronchial asthma associated with allergy to fur allergens.
Tezepelumab	Anti-TSLP antibody	Severe bronchial asthma	Treatment of allergic bronchial asthma associated with allergy to fur allergens.
Lanadelumab	Plasma anti-kallikrein antibody	Hereditary angioedema	X

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
