# Peer review of "Treatment of Allergies to Fur Animals"

_ijms, 2024, doi:10.3390/ijms25137218_

Round 1

Reviewer 1 Report

Comments and Suggestions for Authors

The purpose of the review by Rosada et al is to give an overview of the treatment of allergies to furry animals. Although the section on specific immunotherapy covers the subject rather well, other sections are too general and not focused enough on animal allergy.

The introduction mixes all forms of allergic exposure to animal allergens and symptoms. A separation of the respiratory exposure from oral exposure would be preferred. Anaphylactic shock is related to food or animal bites and should not be listed with respiratory exposure, which can lead to asthma exacerbation. Symptoms of local allergic rhinitis are described in a large paragraph. How frequent is this condition? A precise diagnosis is crucial before doing any specific immunotherapy. However, a section on molecular diagnosis is completely lacking. Figure 1 lists all animal allergens, but does not give any details on their availability and use in molecular diagnosis.

For the section on allergy prevention and allergen elimination, the consensus document by Davila et al could be referenced.

The section of allergy treatment, including table 1, is very general and not focused on animal allergy. Regarding AIT, there are no guidelines which therapy should be preferentially used for which patient. e.g. uncontrolled asthma is an exclusion criteria for AIT. Table 3 lists biological drugs, but again, the section is not specifically addressing allergy to furry animals.  

Introduction lines 24-31: the cited references relate on allergic sensitization, not allergy. Please correct. The European study probably refers to the Ga2len study. Please add the original reference here.

Figure 1: Fel d 1 and Ory c 3 are both secretoglobins, please correct. Lipophillins belong to the secretoglobin family. Ref 4 and 5 seem to be identical. As they are not in English language, I suggest to also add the reference 3 (Dramburg et al) to the legend of the figure.

Figure 2: what is the meaning of ‘allergen’ and ‘specific’ ? How does allergen-specific IT fit into this scheme?

Line 62: ‘anaphylactic shock after exposure to animal allergen’ suggest to add ‘animal food allergens’ to distinguish from respiratory exposure. Anaphylactic shock can also occur from bites of mice and rats. Anaphylactic shock to animal bites is reviewed in a recent study by G Stave, 2023.

Line 72: authors emphasize that diagnosis is not easy, but they do not give an explanation why (IgE cross-reactivities) and how a precise diagnosis could be achieved by molecular allergology.

Line 107: it’s not the fur that elicits symptoms, but the allergens sticking to the cloth. In particular, this reviewer is not aware of a situation of anaphylactic shock related to contact with contaminated cloth. Please give a reference.

Line 118: please give a reference for the statement of the 100 times increased dose used in AIT.

Table 2: are all those products licensed for AIT In Poland? Are they also available in other EU countries?

Comments on the Quality of English Language

Minor editing needed

Author Response

Dear Reviewer 1,

Thank you for your interest in our Manuscript and the time and effort it took to review it. We have now revised the article and we hope that the sections, that you have found too general and not focused, are now more clear, interesting and informative.

In the introduction all forms of allergic exposure to animal allergens and symptoms are now separated, especially respiratory exposure and oral exposure. The frequency of local allergic rhinitis was added.

The article is focused mainly on the treatment of furry animals allergy. Recently we published a detailed description of the available diagnostic methods in animal allergy. The allergen components available in different methods are described there in detailed. We referenced this article in an appropriate place.

We also referenced important consensus documents, as requested.

The guidelines, indications and contraindications to AIT in animal allergy are now clearly explained.

“Sensitization” was mixed with “allergy” in the introduction – we corrected it as well. Secretoglobulins were pointed out in Fig 1.

All references were corrected and updated. Figure 2 was corrected – there were information in this figure, that were lost during editing,

The exposition to animal fur on clothes might cause symptoms in patients – in a form more of an asthma attack, then anaphylaxis – it was modified.

The dose of allergen used in AIT is much higher then regular exposition – the article referenced in our Manuscript shown that it is in fact 100x higher, but we changed it to much more safe – “higher”, without actual the number.

It is difficult to find a list of AIT products available for the EU countries. But we did our best.

We hope that you will find all the corrections adequate, necessary and up to date. Thank you for your input and for helping us make our work better. We feel that it improved a lot and hopefully you will find it ready for publication.

Kind regards

Tomasz Rosada, on behalf of the authors

Reviewer 2 Report

Comments and Suggestions for Authors

The review by Rosada and coauthors is nicely written and concise overview of currently available treatment options for allergy to furry animals. However, there are some points that could be improved.

1. The authors primarily focus on allergy to cats and dogs. More results obtained for other fur animals should be added to the review.

2. The chapter with more details on most common allergens would be good addition to the review (biochemical properties as well as more clinical data on prevalence of anti-IgE antibodies to different proteins).

3. A recent review paper (10.5414/ALX02454E) by Hinger et al. discusses hypoallergenic animals and should be cited as it is highly relevant.

4. In several instances authors focus on the state of matter in Poland, listing allergens extracts and drugs available in Poland. The authors should, therefore, emphasize regional aspect of their review in the abstract and introduction.

5. Some technical issues that I noticed:

-          Affiliation number 2 is duplicated and corresponding author is not labeled with asterisk.

-          First raw in Table 1 should be formatted so that words such as „horse“, „European“ are in one line. Maybe it would be better if the Table is in landscape format?

-          In Figure 2, „treatment of allergy“ instead of „treatment of allergic“. Also, why is „specific“ divided into „allergen“ and „specific“?

-          References 4 and 5 in reference list are duplicated.

Author Response

Dear Reviewer 2,

Thank you for your kind review. It is a pleasure to know that you have found our Manuscript nicely written and concise.

As you notices most information in our work is focused on cat and dog allergy. The simple explanation is, that there are only few articles published up until today, that focused on other furry animal allergy. We put a lot of effort to show current knowledge on as many furry animals as possible. We published before articles that focus only on allergen properties of furry animals and another paper on diagnosis of pet sensitization. The goal of this article is to show treatment options that are available currently.

Hypoallergenic animals is in fact an interesting subject and it was added. Thank you for pointing it out.

We modified the Manuscript, for it to be more focused on the current situation in EU, not only Poland.

Technical issues, including mistakes in Fig 2 were corrected.

We hope, that in the current form the Manuscript is now much more understandable, interesting and complete. Thank you for your help in making our work better.

Kind regards

Tomasz Rosada, on behalf of the authors

Round 2

Reviewer 1 Report

Comments and Suggestions for Authors

The manuscript now reads very well, it is more focused and more relevant to the public interested in this specific topic.

Figure 1 still shows Fel d 1 and Ory c 3 as 2 categories. However, they both belong to the secretoglobin protein family. Lipophillins are not a protein family.

Equ c 1,is marked with a star, but it is a lipocalin.

Comments on the Quality of English Language

no comments

Author Response

Dear reviewer,

thanks for your valuable comments. Changes have been made according to your suggestions.

Kind regards,

Tomasz Rosada

Reviewer 2 Report

Comments and Suggestions for Authors

The authors have made substantial improvements to the manuscript.

Author Response

Dear reviewer,

thanks for your valuable comments.

Tomasz Rosada